# A Mysterious Health Crisis in Aswan Governorate, Southern Egypt, September 2024: A Case Report

**DOI:** 10.3390/microorganisms13040785

**Published:** 2025-03-29

**Authors:** Marwa Omar, Heba Abdelal

**Affiliations:** 1Department of Medical Parasitology, Faculty of Medicine, Zagazig University, Zagazig 44519, Egypt; 2LIS Cross-National Data Center, Maison des Sciences Humaines, L-4366 Esch-sur-Alzette, Luxembourg

**Keywords:** Aswan, *Escherichia coli*, outbreak, health crisis, Egypt, water pollution

## Abstract

In September 2024, the Egyptian Health Ministry declared an *Escherichia coli* (*E. coli*) outbreak in the southern province of Aswan. The spread of an ambiguous illness erupted in the village of Abu Al-Rish Bahri, 20 kilometers north of Aswan, with hundreds of citizens arriving at the governorate’s local hospitals suffering from severe gastrointestinal infections. The authorities, however, did not trace the outbreak’s most common source nor determine whether it was food- or water-borne. The official explanations for the frequent cases and the reported fatalities were inconclusive. There was an evident lack of comprehensive documentation on the extent of the infection, the exposed population, the prevalence pattern of the pathogen, or the retrieved *E. coli* isolates. In addition, the Egyptian government denied any possible association between the contamination of drinking water and the recent Aswan crisis. Challenging the official narrative, this article proposes a scientific report based on featuring the status of *E. coli* infection in Egypt, highlighting the gaps in the announced outbreak claims and adapting water pollution as an alarming hypothesis for the peculiar Aswan disease.

## 1. Introduction

Aswan is a famous historical destination in the south of Egypt. The governorate is located 890 km (553 miles) to the south of Cairo. It is the third most popular place to visit in Egypt, after Cairo and Luxor. Aswan covers a total area of 2807.8 km^2^, with 62% of its area being plain and the remaining 38% being mountainous. In 2006, the total population of the Aswan metropolitan region was around 1.2 million [1].

On 11 September 2024, a mysterious health crisis erupted with the arrival of hundreds of Aswan citizens suffering from severe gastrointestinal infections at the governorate’s local hospitals. The alarming spread of an ambiguous illness was initially recorded among the residents of Abu Al-Rish Bahri village and the nearby areas surrounding the Daraw Centre, 20 km north of Aswan. Three days later, on 14 September, reports of similar cases were documented in Wadi Al Arab village in Nasr Al Nuba district (50 km away from the Daraw Centre) [2,3].

The citizens presented to the local hospitals with severe watery diarrhea that lasted for four days, nausea, vomiting, and abdominal pain. Some patients experienced serious deterioration in their health condition, leading to severe dehydration and damage to kidney functions. The provisional diagnosis was acute gastroenteritis. However, the direct etiology of the disease was yet to be determined. According to the local health authorities, Aswan hospitals received around 480 cases of gastroenteritis since 11 September. Most of the admitted cases were mild, except for 168 patients, who required hospitalization. Of those, 49 were cured and discharged, while 78 were recovering. The Egyptian Health Minister claimed that 36 individuals with pre-existing chronic conditions remained in intensive care units [3].

The fact that so many people fell sick at the same time suggests that something ominous was at play in the southern province. However, the Egyptian authorities downplayed the seriousness of the situation and pushed for reassurance despite the lack of certainty about the nature of the disease or its causes. Speculations about the escalating crisis in Aswan ranged between the possibility of an outbreak and water pollution.

To better understand the situation in Aswan, the first part of this article presents the official narrative adapted by the Egyptian government. The second part, however, is based on an analysis of *E. coli* infection in different Egyptian localities, outlining the knowledge gaps in the reported outbreak statements and introducing drinking water pollution as a plausible explanation for the crisis.

## 2. The Narrative of the Egyptian Government: Confusion and Denial

Denial has been the dominant theme in the statements issued by the Egyptian government, with conflicting accounts on the source of the outbreak, number, and causes of fatalities.

### 2.1. Source of Infection

In his first official comment on the crisis, the Egyptian Health Ministry spokesman denied the presence of any specific bacteria in the water or the food samples collected from street vendors in Aswan [4]. Another hypothesis was proposed by a representative of Aswan Governorate in the House of Representatives, who pointed out that the crisis coincided with the celebrations of a religious ceremony in Egypt, during which sweets are traditionally sold. The official claimed the real reason behind the outbreak was the consumption of local sweets during the ceremony [5]. However, the relatives of some patients denied any history of purchasing such sweets or attending any festivities [3]. In contrast, the Egyptian Health Minister attributed the infections to contaminated food or drink during a press conference held on 23 September at Al-Sadaka Hospital in Aswan. The minister incriminated *Escherichia coli* (*E. coli*) bacteria as the source of the recent outbreak [6].

### 2.2. The Death Toll in Aswan

On 21 September, a member of the House of Representatives for Aswan Governorate announced the sudden death of seven cases, six of which died in one day, raising alarming concerns over the exact causes of such severe acute infections [7]. Conversely, the Egyptian President’s advisor for health and prevention affairs confirmed on 24 September that the infection in Aswan resulted in the deaths of five individuals suffering from chronic diseases. The Egyptian official further denied any direct relationship between (*E. coli*) infection and the reported deaths [8].

On 22 September, the Aswan Governor attributed the causes of death to heart failure or diseases other than intestinal infections. Such a statement was abrogated by an official death certificate issued by Aswan University Hospital on 19 September. The document described the death of a resident after complete circulatory, respiratory, and cardiac arrests due to severe gastrointestinal-related complications, including dehydration and impaired kidney function, according to a report published by the Egyptian journal Mada Masr [9].

### 2.3. Drinking Water Safety

The Egyptian authorities disregarded the hypothesis of water pollution in Aswan. The Ministry of Health (MOH) defended the safety of samples collected from 103 water stations in the Upper Egyptian Governorate and confirmed the absence of any microbiological or chemical changes [10]. Additionally, the Aswan Drinking Water and Sanitation Company, owned by the government, claimed that the chlorine percentage and concentration complied with specifications. Yet the governor of Aswan backed the assertion regarding the lack of necessary drinking water treatment in the southern province, promising an update of the purification system at Sheikh Ali station in Abu Al-Rish village with the latest methods of chlorine pumping to meet health safety standards [3].

## 3. Scientific Report: Challenging the Official Narrative

### 3.1. Escherichia coli: A Versatile Pathogen with a Ubiquitous Nature

*Escherichia coli* (*E. coli*) is a facultative, anaerobic Gram-negative bacillus of the family *Enterobacteriaceae*, falling under the *Escherichia* genus. Around 90% of *E. coli* strains are commensals inhabiting the intestinal tracts of humans and warm-blooded animals. As a commensal, the bacterium lives in a mutually beneficial relationship with its hosts and rarely causes diseases. However, in immunosuppressed patients or in healthy individuals whose physical, anatomical, and physiological barriers have been compromised, *E. coli* can cause severe pathologies, ranging from intestinal infections, such as diarrhea and dysentery, to invasive extraintestinal complications, including bacteremia and sepsis [11].

Most *E. coli* found in the environment are non-pathogenic. However, their presence in food or water can signal inadequate cleaning and careless handling. The pathogenic (*E. coli*) strains differ from their commensal counterparts in their potential to encode specific virulence traits that render them capable of causing the disease. At least 11 (*E. coli*) pathotypes have been defined. Based on the type of virulence factors present and the host’s clinical symptoms, these strains are broadly divided into two categories, intestinal pathogenic *E. coli* (InPEC) and extraintestinal pathogenic *E. coli* (ExPEC) [12]. The intestinal pathogenesis caused by *E. coli* strains can be further subdivided into at least six groups, such as enteropathogenic *E. coli* (EPEC), enterohaemorrhagic *E. coli* (EHEC) or Shiga toxin-producing *E. coli* (STEC), enterotoxigenic *E. coli* (ETEC), enteroinvasive *E. coli* (EIEC), enteroaggregative *E. coli* (EAEC), and diffusely adherent *E. coli* (DAEC). On the other hand, the strains causing extraintestinal infections (ExPEC) include the following pathotypes: uropathogenic *E. coli* (UPEC), neonatal meningitis *E. coli* (NMEC), and necrotoxigenic *E. coli* (NTEC) [11].

### 3.2. An Overview of the (E. coli) Situation in Egypt

No *E. coli* outbreaks were reported in Egypt prior to the recent announcement in September 2024. Nonetheless, numerous studies have investigated the prevalence and pathogenic potential of *E. coli* in various food supplies, water sources, and human or environmental samples across different Egyptian localities. To the north of the country, in the region of the Nile Delta, on the eastern side, around 32 strains of *E. coli* were identified in different meat products (ground beef, beef sausage, beef burger, and beef luncheon) sold in Mansoura city of El-Dakahlia Governorate. Among the isolated *E. coli* strains, 37.5% were potentially diarrheagenic [13]. Furthermore, the possibility of *E. coli* contamination of drinking water samples was explored in Sharkia Governorate. An overall *E. coli* prevalence of 5.33% was recorded. The diarrheagenic *E*. *coli* (DEC) virulence genes, Intimin (*eae A*), Shiga toxin 1 (*stx1*), and Shiga toxin 2 (*stx2*) were detected in 33.33%, 20.0%, and 6.66% of the tested tap water samples, respectively [14].

In the south of the Delta, in Menofia Governorate, the local dairy products (raw milk and raw milk-derived cheese, Karish and Ras cheese) were tested for the risk of pathogenic *E. coli* transmission. It has been reported that 36.9% of the examined dairy products carried potentially virulent *E. coli* genes [15]. In the mid-Delta region of Kafrelsheikh Governorate, contamination of the farm environment with *E. coli* (STEC) isolates was confirmed in 12% of diarrheic cattle samples [16]. Additionally, dissemination of *E. coli* to the aquatic niche has been investigated in the governorate of El Beheira in the western Delta, where 40 *E. coli* isolates were identified in the fresh Nile tilapia fish. The virulent (*eaeA*) gene was detected in 83.3% of the tested isolates [17].

In Cairo, the Egyptian capital, *E. coli* was isolated from stool samples of Egyptian children with diarrhea. Of the examined DEC pathotypes, EAEC was recovered in 30.7% of the diarrheic samples, while STEC and EIEC strains were not detected [18].

Despite some reports on the prevalence of *E. coli* in the Upper Egyptian Governorate of Aswan, the extent of the associated risk factors is not well defined. Different meat products, randomly collected from several shops in the governorate, were bacteriologically tested for STEC serotypes. Raw meat samples had higher infection rates for both *E. coli* O157:H7 and non-O157:H7 *E. coli* than the processed ones [19]. Such a finding has recently been supported following the isolation of *E. coli* at a higher rate in fresh (40%) than minced (20%) meat [20]. In 2021, an overall *E. coli* prevalence of (20.83%) was recorded for ready-to-eat meat products (sausage, hamburger, minced beef and fried chicken) distributed in the markets of Aswan City [21].

### 3.3. Knowledge Gaps in the Official Outbreak Claims

Based on the aforementioned governmental narrative, the official claims regarding the late Aswan outbreak remain vague and elusive. So far, there is an evident shortage of data on the local source of infection, transmission routes, epidemiological variables, demographic characteristics of patients, or the food chain directly linked to the outbreak. Moreover, the results of the outbreak epidemiological and microbiological investigations were inconclusive. No reliable data were available on the magnitude of infection, the distribution of specific *E. coli* pathotypes, or the prevalence of virulence genes among the recovered *E. coli* isolates.

### 3.4. Is Water to Blame?

Despite the persistent negation of the official parties [10], the possibility of drinking contaminated water as a primary cause of the recent Aswan crisis should not be excluded. Since the outbreak was reported at the level of villages (Abu Al-Rish and Wadi Al Arab), it seems plausible that water is a common factor among the documented cases. The infections would have been limited to a single family if food supplies were incriminated.

One of the alarming events that occurred only one month before the crisis was the temporary shutdown of the drinking water stations in the same areas where the infections initially erupted. In August 2024, the Aswan Drinking Water and Sanitation Company announced a temporary shutdown of Sheikh Ali and Abu Al-Rish drinking water stations following the flow of large quantities of turbid water that the stations did not expect. Such pollution had resulted in severe intestinal injuries among the citizens [22].

#### 3.4.1. Water Pollution in Aswan: An Old Crisis

The Nile River is Egypt’s primary freshwater source. The quality of the Nile water depends on population density, the availability of sanitation systems, the extent of industrialization, and economic conditions. The Nile extends for about 1600 km in Egypt, and its watercourse receives waste discharges from 264 different sources, of which 121 are agricultural drains, 70 are industrial outfalls, and 73 are sewerage discharges [23].

EL-Sail or (Kima) Drain is regarded as one of the chief sources of Nile pollution in Aswan Governorate. The drain, which extends about 9 km to the north of Aswan City, was initially constructed to protect the city from temporary floods. Yet, nowadays, it has become a serious human and environmental hazard [24]. EL-Sail Drain is used for the disposal of either treated or untreated sewage, as well as domestic and agricultural wastewater. It also receives the industrial discharge of Kima Company (San Francisco, CA, USA), a large organization in the nitrogen fertilizer industry. The company was built in 1956 and is located about 5.0 km northeast of the Old Aswan Dam to the east of the Nile River [25]. The industrial wastewater effluents of the Kima factory have grave impacts on the water quality in El-Sail stream, which dumps about (56–70 km^3^/day) of its polluted water directly into the Nile [24,26].

#### 3.4.2. Pollution Inputs from El-Sail Drain

The alarming impacts of El-Sail Drain on the quality of the Nile’s water have been thoroughly investigated. In 2015, the drain recorded high microbial loads of the pathogenic bacteria *Salmonella* spp., *Shigella* spp., and *E. coli*, which were recovered in more than 97% of drain water samples [24]. The poor water quality of El-Sail Drain was further confirmed in 2021 in a study reporting that the area around the drain had the lowest water quality index (WQI), exhibiting the highest values of bacterial indicators of sewage pollution (total coliforms, fecal coliforms, and fecal streptococci) [27].

Testing of the other drain water parameters revealed considerable variations in the physicochemical characteristics of the Nile water in the vicinity of El-Sail Drain, which recorded high levels of turbidity and total dissolved solids (TDSs) [28]. The drain water concentrations of different heavy metals, particularly lead (Pb), cadmium (Cd), and nickel (Ni), exceeded the maximum permissible limits according to the Egyptian Nile Protection Law No. 48/1982 [29]. Moreover, the nitrogen-containing compounds (nitrate NO_3_, nitrite NO_2_, and ammonia NH_3_) were among the most significant pollutants recovered from El-Sail Drain. This can be attributed to the Kima Company’s production of ammonium nitrate coated with limestone powder fertilizer. The company spills massive quantities of its untreated industrial waste into the drain, leading it directly to the Nile [26,27,28].

#### 3.4.3. Impacts of El-Sail Drain on the Quality of Abu Al-Rish Drinking Water Station

While point source pollution like El-Sail Drain originates from a specific place, it can affect miles of waterways. The high volumes of household and industrial discharges augment the concentrations of heavy metals and other contaminants entering the receiving watercourse [30]. As a result, the negative impacts of the drain could potentially extend to the nearby drinking water stations [28].

Abu Al-Rish drinking water station is located at a distance of 2.4 km from El-Sail Drain intersection point, in the direction of the Nile flow. The water content of the station is constantly affected by loads of organic and inorganic waste from the drain [24]. Water samples collected from the station exhibited high turbidity levels and significant records of manganese (Mn^2+^) and iron (Fe^2+^) heavy metals, which exceeded the standard values [26]. Heavy metals are regarded as the most active pollutants in the aquatic environment due to their environmental persistence and long-term impacts on living organisms, since some metals tend to accumulate in different organs of humans and animals, posing serious public and environmental risks [31]. In addition, significant levels of NO_2_, NO_3_, and NH_3_ were reported in Abu Al-Rish water station. The neutral, unionized form (NH_3_) is highly toxic to plants, fish, and other aquatic elements [26].

#### 3.4.4. Governmental Disregard for a Court Ruling Banning Nile Pollution

In February 2021, the Administrative Court of Aswan issued a ruling that mandated the government to take immediate measures to prevent the discharge of wastewater and industrial pollutants into El-Sail Drain. However, the government blatantly appealed the ruling in August 2022, endangering thousands of lives. The court conducted a detailed field inspection of El-Sail Drain, spanning 8 km from south to north. To ensure accuracy, the court rejected the results submitted by the Ministry of Health’s laboratories, as they were unreliable. They contradicted the analyses performed by the Ministry of Irrigation and the Environmental Affairs Agency, which confirmed the presence of untreated industrial waste dumped directly into the Nile through El-Sail Drain [32].

## 4. Conclusions and Recommendations

The recent health crisis reported in Aswan Governorate, Southern Egypt, has revealed a lack of transparency from the Egyptian government and a loss of credibility in its official statements. The Egyptian authorities failed to provide grounded explanations for the rising cases of an alarming illness in the southern province. The results of the outbreak investigations were inconclusive. No detailed information was available on the extent of the bacteria responsible for the infections or the specific outbreak strain. Therefore, we ought to look beyond the scope of confining the situation in Aswan solely to *E. coli* infections. The real reasons behind the crisis should be uncovered.

Despite the official assurance about the safety and readiness of drinking water in Aswan, the Abu Al-Rish water filtration station, located at a distance of 2.4 km from El-Sail stream, is constantly exposed to high loads of toxins from the drain, endangering the lives of Aswan residents. In response to these concerns, it is urgent to call for the following:*General recommendations:*
1.Thorough outbreak investigations as part of prompt responses to a public health emergency.2.Generation of epidemiological data on the risk factors linked to food- or water-borne infections.3.Employment of advanced molecular-level characterization techniques to tackle health challenges in resource-limited settings.
*Specific recommendations:*
1.Improvement of water infrastructure and the sewage disposal system in the village of Abu Al-Rish and nearby areas to prevent the mixing of drinking water with sewage.2.Updating the chlorine system of Abu Al-Rish and Sheikh Ali drinking water stations to meet health safety standards.3.Relocation of the main water intake of the infected villages to a safer place away from the pollution source.4.Regular environmental and occupational health monitoring of the drinking water stations in Aswan Governorate.5.The local official parties must scale up massive community intervention measures to combat the outbreak, necessitating disinfection of the environment, isolation of patients, and management of contacts.6.Presentation of the results of investigations, which should be made public to ensure transparency.7.Post-outbreak surveillance and assessment of the drinking water quality in the infected villages.8.Enforcement of a court ruling that mandates the Egyptian government to stop dumping toxins and pollutants into the Nile River.9.Implementation of immediate coordinated measures for the treatment of different wastes discharged directly into Nile water or leaching into El-Sail Drain, which contributes substantial pollution load to the river.

## Data Availability

All data generated during this study are included within this article.

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
