# Peer review of "A Mysterious Health Crisis in Aswan Governorate, Southern Egypt, September 2024: A Case Report"

_microorganisms, 2025, doi:10.3390/microorganisms13040785_

Round 1
Reviewer 1 Report
Comments and Suggestions for Authors
This manuscript aimed to provide a scientific opinion on the mysterious health crisis that occurred in September 2024 in the Aswan Governorate, Southern Egypt. It is important to note that it is presented as a scientific opinion piece rather than as a traditional research study with a formal methodology. Based on the collected information, the authors propose a plausible alternative explanation for the health crisis, suggesting that water pollution is a potential cause. However, it does not involve primary data collection or analysis specific to the September 2024 outbreak, and the validity of the approach has not been formally evaluated or tested in a traditional scientific sense. In addition, a comprehensive epidemiological analysis of the outbreak was not provided, which is crucial for determining its cause. Therefore, this manuscript is not suitable for publication in Microorganisms.
Reviewer 2 Report
Comments and Suggestions for Authors
The article delves into the recent E. coli outbreak in Egypt, with a focus on the implications of environmental pollution, on public health. It discusses the pathogenic potential of various E. coli strains found in food and water sources. The article effectively outlines the public health risks associated with E. coli, other bacterial infections and heavy metals in Egypt, driven by environmental pollution from the El-Sail Drain. The thesis emphasizes the necessity for governmental accountability and intervention to protect public health. Overall, the article serves as a crucial reminder of the intersection between environmental factors and health crises, urging immediate action to safeguard the wellbeing of the affected populations.
The urgency of the recommendations is clear. There is a strong call for both immediate action (e.g., improving water quality, controlling pollution) and strategic long-term solutions (e.g., improving water network, scaling up community programs). This helps underscore the gravity of the situation.
The authors could include more information from affected communities to provide a more defined picture of the crisis. Additionally, discussing potential solutions or strategies for engaging the community in prevention efforts could strengthen the call to action.
The implementation details for certain recommendations need to be more specific, and the tone could be adjusted for better engagement with all stakeholders. A revision that addresses these aspects would make the conclusions and recommendations more actionable and impactful.
Reviewer 3 Report
Comments and Suggestions for Authors
The manuscript details a diarrheal episode in Aswan Province, southern Egypt, in September 2024, affecting a significant portion of the population and outlining the government health sector's response. The authors effectively highlight the challenges in diagnosis and the limited measures implemented by the authorities. However, the manuscript lacks clear framing as an opinion piece, reading more as a descriptive account of a local event or a critical scientific report. What is missing is a stronger, more assertive scientific opinion. No scientific data is discussed for or against the episode being an outbreak.
Round 2
Reviewer 1 Report
Comments and Suggestions for Authors
The manuscript does not fit any type of article published in Microorganisms.
Reviewer 2 Report
Comments and Suggestions for Authors
Thank you for addressing my comments and suggestions. The article now is more informative than the first version.